# Magnetic Resonance Imaging as a Prognostic Disability Marker in Clinically Isolated Syndrome and Multiple Sclerosis: A Systematic Review and Meta-Analysis

**DOI:** 10.3390/diagnostics12020270

**Published:** 2022-01-21

**Authors:** Amjad I. AlTokhis, Abrar AlAmrani, Abdulmajeed Alotaibi, Anna Podlasek, Cris S. Constantinescu

**Affiliations:** 1Mental and Clinical Neuroscience Academic Unit, School of Medicine, Nottingham University Hospitals NHS Trust, University of Nottingham, Nottingham NG7 2UH, UK; abdulmajeed.alotaibi@nottingham.ac.uk (A.A.); anna.podlasek@nottingham.ac.uk (A.P.); 2Division of Health and Rehabilitation Sciences, Princess Nourah Bint Abdulrahman University, Riyadh 11564, Saudi Arabia; 3Faculty of Health, York University, Toronto, ON M3J 1P3, Canada; abrar92@yorku.ca; 4Department of Radiological Sciences, School of Applied Medical Sciences, King Saud bin Abdul-Aziz University for Health Sciences, Riyadh 14611, Saudi Arabia; 5Tayside Innovation MedTech Ecosystem, Division of Imaging Science and Technology, School of Medicine, University of Dundee, Dundee DD1 4HN, UK; 6Department of Neurology, Cooper Neurological Institute, Camden, NJ 08103, USA; c.s.constantinescu@gmail.com

**Keywords:** MRI, multiple sclerosis, meta-analysis, prognostic markers, white matter lesions

## Abstract

To date, there are no definite imaging predictors for long-term disability in multiple sclerosis (MS). Magnetic resonance imaging (MRI) is the key prognostic tool for MS, primarily at the early stage of the disease. Recent findings showed that white matter lesion (WML) counts and volumes could predict long-term disability for MS. However, the prognostic value of MRI in the early stage of the disease and its link to long-term physical disability have not been assessed systematically and quantitatively. A meta-analysis was conducted using studies from four databases to assess whether MS lesion counts and volumes at baseline MRI scans could predict long-term disability, assessed by the expanded disability status scale (EDSS). Fifteen studies were eligible for the qualitative analysis and three studies for meta-analysis. T2 brain lesion counts and volumes after the disease onset were associated with disability progression after 10 years. Four or more lesions at baseline showed a highly significant association with EDSS 3 and EDSS 6, with a pooled OR of 4.10 and 4.3, respectively. The risk increased when more than 10 lesions were present. This review and meta-analysis confirmed that lesion counts and volumes could be associated with disability and might offer additional valid guidance in treatment decision making. Future work is essential to determine whether these prognostic markers have high predictive potential.

## 1. Introduction

Magnetic resonance imaging (MRI) has an essential role in multiple sclerosis (MS) diagnosis and prognosis, for estimating the risk of developing MS [1] and assessing MS disease activity when making treatment decisions [2,3,4]. The neurological disability varies among patients, from practically asymptomatic patients [5,6] to those with severe neurological disability and shortened life. Recently, early treatment initiation has gained an interest for preventing/eliminating long-term disability and induce remission. However, disease-modifying therapy (DMT) might be associated with life-changing side effects [7,8].

The first presentation of MS is referred to as a clinically isolated syndrome (CIS). Patients present with an episode of neurological symptoms that at least partially resolves [9]. However, up to 85% of CIS patients develop relapsing-remitting MS (RRMS) [9]. Following a CIS, a higher initial brain white matter lesion (WML) load and volume over the first 5 years were thought to be associated with the increasing risk of developing the disability and conversion MS subtype in 20 years [10,11]. Given this, assessing the association between WML lesion load and volume on an MRI scan obtained at the initial presentation (CIS) and their prediction of long-term disability is crucial [12].

Different reviews have been performed on MRI factors and their association with the risk of developing MS [13,14]; however, most do not fulfil the systematic review criteria [15,16,17]. To our knowledge, this is the first systematic review with a meta-analysis assessing the prognostic value of MRI in MS and predicting long-term physical disability.

## 2. Materials and Methods

### 2.1. Study Registration

This systematic review and meta-analysis were conducted according to the preferred reporting items for systematic reviews and meta-analysis (updated PRISMA) guidelines [16] and were registered in the international prospective register of systematic reviews (PROSPERO) database (CRD42021249236) in May 2021.

### 2.2. Sources, Search Strategy, and Screening

Four electronic databases were searched (Medline, Embase, Web of Science, and PubMed) with the keywords (“Magnetic Resonance Imaging” or “MRI”), (“Lesion” or “lesions”), (“Count or Counts”), and (“Multiple Sclerosis” or “MS”). For the Medline search strategy, see Appendix A. Mendeley was used for bibliographic management, including duplicate removal.

The search strategy was conducted by three reviewers A.I.A. (Amjad I. AlTokhis), A.A. (Abrar AlAmrani), and A.A. (Abdulmajeed AlOtaibi), who also verified the database selections, study screening/identification, study eligibility/inclusion, and quality assessments independently and blindly.

### 2.3. Study Selection (Inclusion and Exclusion Criteria)

Studies were considered for inclusion if they: (i) were longitudinal prospective and retrospective studies; (ii) reported the association between baseline MRI lesion count or volume and MS disability; (iii) included ≥ 10 years of follow-ups to assess disease progression, as disability is usually a slow process from the disease onset, and the predictive value of the short-term disability evolution within the first years is low [18]; (iv) investigated the association between white matter lesions count/volume in MS and disability, with at least one of the MRI measurements (T1/T2 lesion number or count) at the baseline; (v) reported EDSS as an outcome; (vi) were published in peer-reviewed journals; (vii) were written in English; and (viii) included all MS subtypes, but only findings related to early relapse onset (CIS) were included.

Studies were considered for exclusion if: (i) they were placebo arms of randomised controlled MS immunotherapy treatment studies, because the placebo arms often convert to treatment after 2 years; (ii) the participants were under 18 years old; (iii) they reported on different MRI techniques (e.g., spectroscopy), or only reported on prognostic models; and/or (iv) they were reviews, opinion pieces, editorials, comments, no-full text articles, technical, or animal studies. A manual search of the reference lists of the included studies was performed to detect further eligible studies.

### 2.4. Data Extraction

In April 2021, full texts were examined independently by three reviewers (A.I.A., A.A., and A.A) according to the predefined inclusion criteria. Disagreement over including or excluding a study was resolved by discussion.

The same reviewers independently extracted demographics, disease-specific data, and outcome data (including age, gender, disease duration, EDSS score, population groups, and sample size), and then the results were compared. All included studies assessing disability, MS, or SPMS in relation to baseline MRI data were extracted.

### 2.5. Outcome Measures

The primary outcome was assessing disability using the expanded disability status scale (EDSS). Thus, studies that evaluated the mean EDSS and time to reach EDSS 3 or 6 were included.

### 2.6. Quality Assessment

To assess the validity of the included studies, the quality in prognosis studies (QIPS) tool [19] was used. It contains six domains to appraise the following: (i) participation; (ii) attrition; (iii) prognostic factor measurement (MRI in this study); (iv) outcome measurement; (v) study confounding; and (vi) analysis. A judgment concerning the related risk of bias was assigned for each domain (yes, partly, no), and a checklist was used to assess the quality and completeness of MRI reporting. Citation chaining from reviews and other papers discovered by reviewers was carried out, and the searches were re-run before the final analysis in June 2021. Disagreement between reviewers was solved by mutual discussion.

### 2.7. Statistical Analysis

Study characteristics and extracted variables were summarised using standard descriptive statistics. Continuous variables were expressed as means and SD, and categorical variables were expressed as frequencies or percentages. The meta-analysis of binary outcomes was expressed as OR with 95% CI. A random-effects model and the Mantel–Haenszel method were used to pool the study the odd ratios and compute an overall *p*-value. Heterogeneity tests were conducted as a chi-square variate (with the assumption of the homogeneity of effect sizes). The I^2^ statistic was used to assess the extent of between-study heterogeneity. Study heterogeneity I^2^ values > 50% were considered to be substantial and those >75% were deemed to have considerable heterogeneity. All *p*-values were two-tailed, with values < 0.05 considered to be statistically significant. Review Manager 5.4.1 software (Cochrane organisation, London, UK) implemented all analyses.

## 3. Results

### 3.1. Literature Search and Study Characteristics

The systematic search retrieved 1108 studies. A total of 423 studies remained after removing duplicate studies and ineligible studies, such as case studies and letters, using an automated tool with each database. Screening the title and abstract excluded 399 studies that did not meet the eligibility criteria. A total of 25 studies remained for full-text assessment. Of those, 15 studies [10,11,12,20,21,22,23,24,25,26,27,28,29,30,31] fulfilled the eligibility criteria for qualitative synthesis, while three studies [10,11,22,23,25,29] were included in the meta-analysis (Figure 1).

### 3.2. Cohort Description

London (The National Hospital, Queen Square; Moorfields Eye Hospital) has two different cohorts, namely London 1 in 1984–1987 and London 2 in 1995–2004 [10,22,23,25]. The London 1 cohort [10,23] included 140 patients scanned at baseline [10] and followed clinically after 5 [32], 10 [22], 14 [25], and 20 years [10] by assessing the EDSS changes throughout the years.

Filippi [33] performed a reanalysis of the 5-years cohort data [32]. Similarly, Sailer [23] and colleagues used the data of 58 patients from the 5- and 10-year London 1 cohort. Filippi and Sailer used semi-automated MRI analysis techniques, and Chung et al. (2020) used the same data after 30 years of follow-ups [12]. In London 2, the cohort was part of a follow-up study of 143 patients and the EDSS was assessed at 5 years.

Barcelona (Vall d’Heborn University Hospital) has a centre-based cohort that started in 1995 and is ongoing [11,34,35]. CIS patients are included consecutively with an assessment of the EDSS and relapses every 3–6 months [11,34,35] or annually [11]. MRI scans were performed at baseline (3–5 months after CIS diagnosis), 12 months after, and then every 5 years [11]. The EDSS was assessed at each visit [11,34,35]. Several studies were published from this cohort, including the first [35], which was a 7-year longitudinal study of 175 patients. In contrast, the study in 2010 focused on the prognostic role of infratentorial lesions at baseline by retrospectively analysing 77 patients with infratentorial lesions out of 246 patients [34]. Another study in 2015 included 1058 patients [11]. A publication [24] in 2019 reported on 401 patients. The cohorts of the included studies used the Poser and McDonald criteria for MS diagnosis.

### 3.3. Quality Assessment

There was high heterogeneity between studies due to several causes, namely, different study designs (centre or multi-centre-based cohort), prospective and retrospective studies, and the broad spectrum of study duration starting from 1984 to ongoing studies (Table 1).

There was no systematic way to report findings, making finding and comparing the results challenging. A striking observation was made of the way the results were reported between the studies; mainly, the raw data related to the primary outcome were not reported and cannot be accessed. For example, the lesion counts and volumes at baseline and follow-up were not reported, making it challenging to compare. Additionally, different findings that were not essential to the primary outcome were reported instead of illustrating that no significant results were found.

Multiple studies were published, but they were using the same cohort [10,11,12,23,24,25,36,37], while other centres did not report the needed information. The Italy cohort [20] were not included in the meta-analysis, as all members of the study cohort reached EDSS 6.

Quality assessment for potential bias showed mixed results. Specifically, quality was not consistent for “prognostic factor” (MRI measurements), “outcome measurement (i.e., EDSS), and “confounding measurement and account” (immunotherapy). Detailed information about essential and raw data was often missing, including lesion count/volume at baseline and their association with EDSS 3 or EDSS 6.

Additionally, variability in MRI sequences, field strengths, and rater-blinding strategies led to more heterogeneity between studies. The MRI data synthesis showed that most of the studies provided information on field strength and scan resolution. However, clinical data blinding was only reported in eight studies, and the number of raters was reported in 11 studies. The most common sequence used in lesion detection was T2-weighted images using a semi-automated tool. Complete information can be found in Appendix A.
diagnostics-12-00270-t001_Table 1Table 1Quality assessment and potential bias: No: 
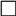
, Partly: 
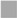
, Yes: 
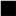
.
Potential Bias (QUIPS) toolStudy, yStudy ParticipationStudy AttritionPrognostic FactorOutcome MeasurementConfounding Measurement and AccountAnalysisTintore, 2020 [24]





Chung, 2020 [12]





Brownlee, 2019 [26]





Tintore, 2015 [11]





Jacobsen, 2014 [38]





Kearney, 2014 [27]





Giorgio, 2014 [29]





Popescu, 2013 [30]





Rovaris, 2011 [31]





Renard, 2010 [20]





Fisniku, 2008 [10]





Chard, 2003 [21]





Brex, 2002 [25]





Sailer, 1998 [23]





O’Riordan, 1998 [22] 








### 3.4. Study Results

At baseline, the patients’ mean age ranged from 29 to 32 years, and 67% were female across all cohorts. However, no data were given regarding the mortality rate after the onset and its correlation with MRI parameters/findings. To predict disability after CIS, we compared different MRI prognostic features, namely, lesion counts and volumes. Additionally, we assessed how these MRI features could also predict the conversion to MS. Limited information about the SPMS transition was reported. Besides disability—which was the main endpoint of this review—the conversion of MS and different prognostic factors were also looked at. Table 2 shows a summary of the included studies, detailing characteristics of the included studies (qualitative and quantitative).

#### 3.4.1. Baseline MRI Lesions

Ten to fourteen years

The London 1 cohort showed a moderate correlation between the number of T2 lesions at baseline and EDSS after a mean of 10 and 14 years of follow-ups (*r* = 0.54, *p* < 0.001) [22] (*r* = 0.47, *p* < 0.001) [25], respectively. In the 14-years follow-up cohort, three time points (5, 10, and 14 years) were studied for both lesion counts and volumes. They followed four MS patients with a normal MRI scan and a median EDSS of 1.75, and 44 MS patients with an abnormal MRI and a median EDSS of 3.5. Additionally, a moderate correlation of newly formed lesions was also reported (*r* = 0.59, *p* < 0.001). With lesion volume, there was a moderate correlation in lesion volumes at 10 years (*r* = 0.48, *p* < 0.001), while a weak correlation was found in the change in lesion volume over 10–14 years of follow-ups (*r* = 0.35, *p* < 0.02) [25]. Baseline lesion counts and volumes seem to be larger and increase over time in patients with worse clinical outcomes [25].

Fifteen to twenty years

In Fisniku, London 1, T2 lesion volume (baseline, 5, 10, 14, and 20 years) correlated moderately after 20 years of follow-ups (*r* = –0.48 to 0.67, *p* < 0.001). The change in T2 lesion volume over 10–14 years (*r* = 0.40, *p* < 0.004) and over 14–20 years (*r* = 0.49, *p* < 0.001) correlated weakly to moderately with the change in EDSS after 20 years [10]. Only one study reported brain atrophy and gadolinium-enhancing lesions as MRI measurements rather than lesion counts and volumes [39].

#### 3.4.2. MS Progression

To capture disease conversion, this was only reported in the London cohort [10,22,23,25]. Fisniku reported that 28 (42%) patients with MS converted to SPMS 20 years later [10]. There was a significant trend for the development of SPMS to be associated with a higher T2 lesion volume increase over 20 years than in patients who remained RRMS (*p* = 0.007), and the T2 lesion volume increase in patients diagnosed with SPMS appeared to be higher than in those with RRMS (*p* = 0.08) [10]. Results on MRI and mortality were not reported.

### 3.5. Meta-Analysis Results

#### 3.5.1. Lesions Count and Disability in MS Patients

Two centres reported on the same cohorts but with different follow-ups, namely, the London cohort [10,12,23,25,36,37] and the Barcelona cohort [11,24]. Thus, for forest plots, we included the study that reported the needed data: London [10], Barcelona [11], and Norway [38]. Renard’s [20] cohort was also excluded from the quantitative analysis, as all their patients reached EDSS 6. Three studies reported on WML count and EDSS 3 and showed a significant association between the increased WML and higher odds of EDSS 3 (Figure 2A). After pooling, four or more lesions at baseline showed a highly significant association with EDSS 3 (*p* < 0.001), with a pooled OR of 4.10 (95% CI 2.73–6.18). The detection of ten or more lesions had even a higher pooled OR of 4.15 (95% CI 2.89–5.95) and was highly significant (*p* < 0.001) (Figure 2B). The heterogeneity was very low for both analyses. A diagram illustrates the key elements of the forest plot in Appendix A.

Three studies reported on WML count and EDSS 6, and all three showed a significant association between the higher number of WML and higher odds of EDSS 6 (Figure 3A). After pooling, four or more lesions at baseline showed a highly significant association with EDSS 6 (*p* < 0.001), with a pooled OR of 4.3 (95% CI 1.094–16.89). The detection of ten or more lesions also showed a high significant association with EDSS 6 (*p* < 0.001), and the pooled OR was 5.54 (95% CI 1.61–19.06) (Figure 3B). The heterogeneity between studies was moderate to fair (I^2^ = 53%, I^2^ = 47% respectively).

#### 3.5.2. Lesion Volume and Disability in MS Patients

Three studies reported on WML volume and disability; however, due to missing raw data, it was impossible to generate forest plots. Thus, descriptive analysis was performed. In the Sailer et al. study, they found a significant correlation at baseline between WML volume and disability (*r =* 0.81, *p* < 0.001), but the change in lesion volume during the first 5 years did not show any association with long-term disability [23]. In contrast, Brex and colleagues found that in CIS patients during the first 5 years, increases in the lesion volume correlated with the degree of long-term disability [25]. This relationship was moderate, so they concluded that the volume of the lesions alone may not be an adequate basis for decisions about the use of DMT. Lastly, in Italy’s cohort, EDSS worsening over 10 years was best correlated with the combination of baseline lesion count and an increasing lesion volume (R = 0.61, *p* < 0.001) [29].

## 4. Discussion

To our knowledge, this is the first systematic review and meta-analysis that assesses the prognostic value of MRI for disability with MS. The results illustrate that few studies assessed the prognostic MRI value beyond 10 years in MS patients. Previous work presented some evidence that baseline lesions are associated with EDSS after 10 years [34]. The larger cohort from Barcelona, which was followed up with after a median of 6.8 years, showed that the risk of an increase in EDSS was higher in patients with 10 or more lesions [11]. The London cohort reported that T2 lesion volume correlated moderately with EDSS after 20 years [10]. Similarly, 14 years of follow-ups showed a correlation between EDSS and lesion count at baseline [25].

These findings are in line with our analysis, which showed that both EDSS 3 and EDSS 6 showed a strong positive association with the accumulation of WML at baseline. Therefore, it can be concluded that lesion number and volume could be predictors for long-term disability with MS. Contrarily, a study with 15 years of follow-ups showed no association between baseline MRI lesion count/volume and long-term physical disability [26]; still, the conversion to another MS subtype was strongly correlated with the baseline MRI.

A longitudinal observational study with a longer follow-up period is needed to capture disease conversion, which was only reported in the London cohort [10,22,23,25].

Data were derived from only three different cohorts in Barcelona, London, and Norway. A quantitative assessment of these data shows a consistent relationship between early brain lesion loads and subsequent disability. All cohorts were restricted to one city, a fact that might bias the findings and not allow generalization to the other centres.

Lesion segmentation and analysis is a vulnerable process, as it depends potentially on the raters’ experience, the software used, the criteria of lesion selection, and image quality. Thus, having a harmonised and standard procedure for lesion segmentation, such as a state-of-the-art automatic segmentation tool (e.g., HyperMapper) [40], could be beneficial for future studies.

In terms of age and sex, young females were more likely to have more WML in an early stage of the disease, as our data have shown that around 70% of the study population were females in their 30s. These results are to be expected, as the peak age of onset is around 30 years [41], and the disease is two to three times more common in females than in males [42]

Two studies reported that spinal cord lesions (≥1 lesion) could predict disability in patients converting to MS [11,39]. However, recent findings highlighted the uncertainty of the prognostic and diagnostic value of spinal cord lesions [43]. Further work is needed to clarify the independent and interaction impact of brain and upper cervical cord atrophy as promising prognostic markers [44].

Although higher brain lesion load could be linked with long-term disability, the correlation between early lesion loads and later disability still only explains a small proportion of disabilities. It is also difficult in clinical practice to utilise a certain threshold of disability, and similarly with disease progression milestones.

### Limitations

There were several limitations to the data that we could access for our review. Relatively few of the included studies were conducted prospectively, such that the others were potentially vulnerable to various biases due to their retrospective design. In addition, the primary study aims varied considerably, and this could contribute to differences in the observed lesion counts and volumes. Treatment information was not available in the publications. Therefore, a potential differential treatment effect on the disability outcome could not be assessed. Few studies were included in the meta-analysis, and this could cause a publication bias whereby only positive findings were published. Additionally, measuring and defining disability is subjective, and EDSS is prone to human error. Lastly, it should be emphasised that subgroup and meta-analysis are by their nature observational, and results must therefore be interpreted with caution.

## 5. Conclusions

This systematic review and meta-analysis have confirmed that the number or the volume of WML detected with MRI has the potential to be a prognostic imaging marker for long-term disability. Nevertheless, there is considerable heterogeneity in their assessment, detection, and reporting across studies. While the meta-analysis presented here highlights their potential for playing a prognostic role in MS, patient and lesion levels may limit sensitivity for initial diagnosis. We believe that there is a need to establish clear criteria for evaluating and reporting WML, which would improve the interpretation and comparison of data acquired across studies. Such criteria would thus allow validation and the potential translation of this imaging marker into future clinical practice. Further work with longer clinical follow-ups is required to determine whether lesion count, volume, and regional brain atrophy can be used as predictive imaging biomarkers in MS.

## Figures and Tables

**Figure 1 diagnostics-12-00270-f001:**
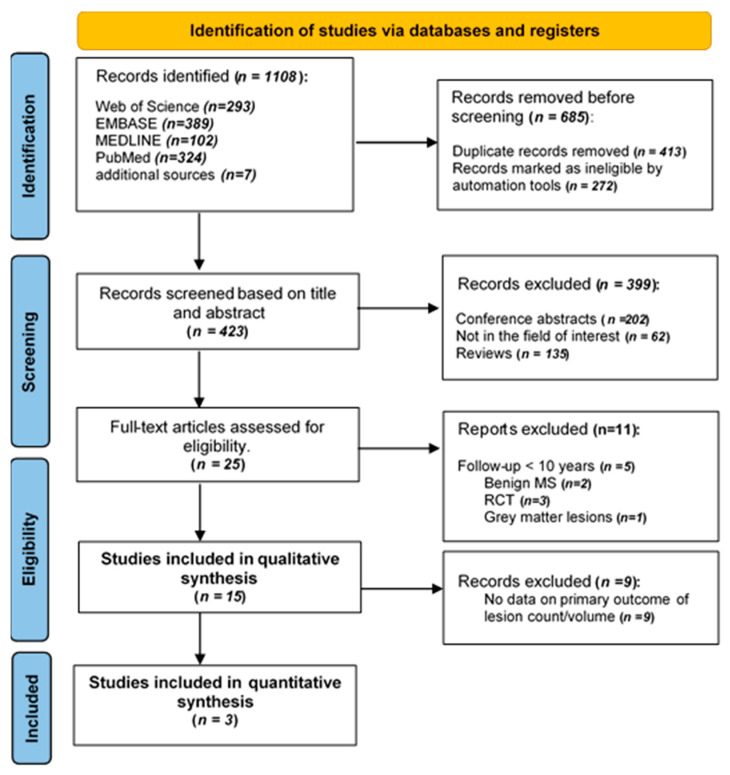
PRISMA flow diagram, illustrating the systematic search strategy and study selection.

**Figure 2 diagnostics-12-00270-f002:**
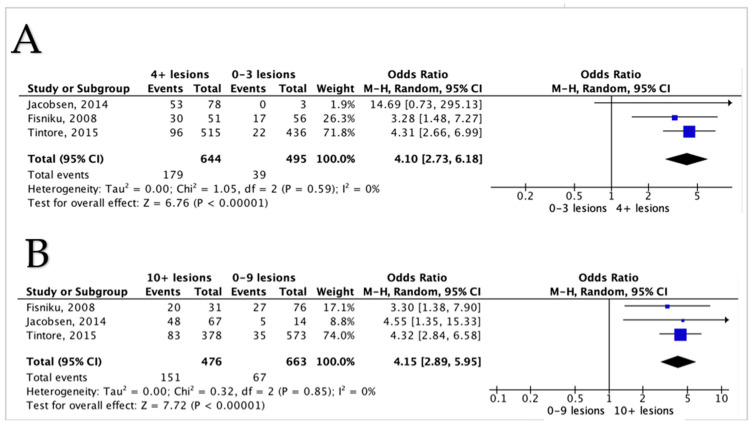
Forest plot demonstrating the odds of EDSS 3 with comparisons between different lesion counts (**A**,**B**); (**A**) a comparison between 0–3 lesions versus 4 or more lesions, (**B**) a comparison between lesions of 0–9 lesions versus 10 or more lesions. CI = confidence interval, I^2^ = heterogeneity index, df = degree of freedom.

**Figure 3 diagnostics-12-00270-f003:**
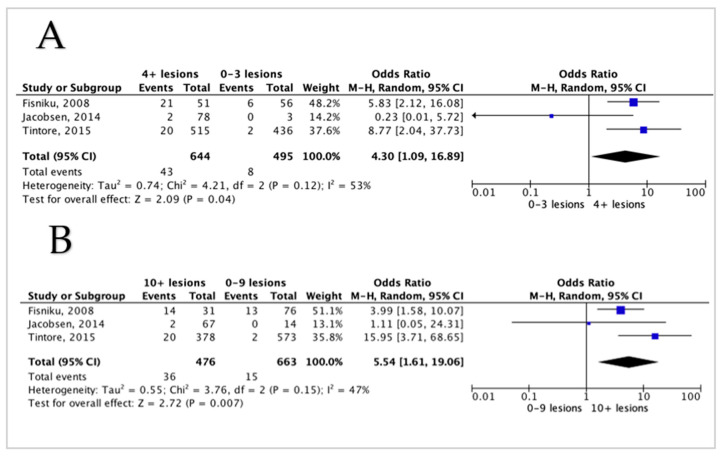
Forest plot demonstrating the odds of EDSS 6 with comparisons between different lesion counts (**A**,**B**); (**A**) a comparison between 0–3 lesions versus 4 or more lesions, (**B**) a comparison between lesions of 0–9 lesions versus 10 or more lesions. CI = confidence interval, I^2^ = heterogeneity index, df = degree of freedom.

**Table 2 diagnostics-12-00270-t002:** Characteristics of the included studies (qualitative and quantitative).

Author, y	Centre and Design	Clinical Features	Study Length	Sample Size	T2 Lesion Count	Lesion Volume	EDSS
Tintore et al. (2020) [24]	Barcelona centre-based prospective cohort 1995–2016	CIS	21 y	401 analysed	0: 80 (20%) 1–3: 67 (17%) 4–9: 56 (14%) ≥10: 188 (48%)	⦸	DMT after CDMS: EDSS = 3 (74 of 156 (47%)) EDSS = 6 (11 of 156 (7%)) DMT before CDMS: EDSS = 3 (11 of 55 (20%)) EDSS = 6 (4 of 55 (7%))
Chung et al. (2020) [12]	UCL, London prospective cohort 1984–1987	CIS	30 y	132 120 analysed	0–16 (86%) 17–20+ (17%)	⦸	⦸
Brownlee et al. (2019) [37]	UCL, London prospective cohort 1995–2004	CIS	15 y	178 164 analysed	⦸	⦸	BL: EDSS = 5 (15.25) FU = 0 (range 0–1) in CIS FU = 2 (range 0–10) in MS
Tintore et al. (2015) [11] *	Barcelona centre-based prospective cohort 1995–2013	CIS	18 y	1058 1015 analysed	0: 299 (31%) 1–3: 137 (14%) 4–9: 137 (14%) ≥10: 378 (40%)	⦸	EDSS > 3 12 of 299 (4%) with 0 les, 10 of 137 (7%) with 1–3 les, 13 of 137 (10%) with 4–9 les, 83 of 378 (22%) with ≥10 les EDSS ≥ 6 2 of 299 (0.7%) with 0 les, 0 (0%) of 137 with 1–3 les, 0 (0%) of 137 with 4–9 les, 20 of 378 (5.3%) with ≥10 les
Jacobsen et al. (2014) [28] *	2 centres in Norway, prospective cohort 1998–2000	MS	10 y	81 analysed	BL: (16.0 ± 12.3)	⦸	EDSS > 3 (50/81) 0 of 50 (0%) with 0 les, 0 of 50 (0%) with 1–3 les, 6 of 50 (12%) with 4–9 les, 44 of 50 (88%), with ≥10 les EDSS ≥ 6 (3/81) 0 of 0 (0%) with 0 les, 0 of 0 (0%) with 1–3 les, 1 of 3 (33.33 %) with 4–9 les, 2 of 3 (6.06 %) with ≥10 les
Kearney et al. (2014) [27]	MAGNIMS (7 centres), retrospective	MS subtypes (CIS, RRMS, SPMS)	26 y	159 analysed	⦸	⦸	EDSS BL:4 (range 0–8)
Giorgio et al. (2014) [29]	Siena, Italy prospective cohort 2000–2001	RRMS	10 y	73 57 analysed	BL: (22.4 ± 18.5) 17 (2–80) New/enlarge: (+1.5 ± 1), 1.3 (0.01 to 4.3)	BL: (5.8 ± 6.4) cm^3^ FU: (8.3 ± 8.1) cm^3^ Annualised 10 y rate of T2 lesion growth (LV change/y) of: (0.25 ± 0.5) cm^3^	EDSS BL: (1.8 ± 1.1) 10 y FU: (2.5 ± 1.7)
Popescu et al. (2013) [30]	MAGNIMS (8 centres), retrospective (before January 2000).	MS subtypes (CIS, RRMS, SPMS, PPMS)	10 y	261 analysed	⦸	BL: 5.91 (2.07–13.82) 1 y FU: 9 (4.2–19) Difference/y 2 (0.5–3.9)	EDSS for the whole group (median (IQR)) BL: (4 (2–6)), 10 y FU: (6 (4–7.5)) EDSS for CIS BL: (0 (0–1)), 10 y FU: (1.5 (1–3)) EDSS for RRMS BL: (2 (1–3)) 10 y, 10 y FU: (3.5 (2–5.5)) EDSS for SPMS BL: (5.5 (4–6)), 10 y FU: (7 (6–7.5)) EDSS for PPMS BL: (5.5 (3.5–6.5)), 10 y FU: (7(6–8))
Rovaris et al. (2011) [31]	MAGNIMS (7centers), retrospective	MS	15 y	369 analysed	⦸	12.4 (0.4–61.1)	EDSS BL: 2 (0–3)
Renard et al. (2010) [20]	3 centres in France, retrospective	RRMS, PPMS	10 y	84 analysed	1–8: 8% 9–20: 12% ≥20: 80%	⦸	EDSS > 6 Of our 84 included patients: 3 had (1–3) les, 4 (4–9) les and 77 had (^3^10) les
Fisniku et al. (2008) [10] *	London centre-based prospective cohort 1984–1987	CIS	20 y	140 107 analysed	⦸	0.43 cm^3^ (median)	EDSS > 3 9 of 34 (26%) with 0 les, 8 of 22 (36%) with 1–3 les, 10 of 20 (50%) with 4–9 les, 20 of 31(65%), with ≥10 les EDSS ≥ 6 2 of 34 (6%) with 0 les, 4 (18%) of 22 with 1–3 les, 7 (35%) of 20, with 4–9 les, 14 (45%) of 31 with ≥10 les
Chard et al., (2003) [21]	London centre-based (1984–1987)	CIS	14 y	28 analysed	⦸	BL: 1 (0.1–3.7) 5 y FU: 2.8 (0–36.6) 10 y FU: 5.8 (0.6–46.1) 14 y FU: 9.4 (1–46.8)	EDSS BL: 2.5 (0–9.5)
Brex et al. (2002) [25]	London centre-based prospective cohort 1984–1987	CIS	14 y	81 71 analysed	⦸	0.43 cm^3^ (median)	EDSS > 3 0 of 21 (0%) with 0 les, 5 of 18 (28%) with 1–3 les, 7 of 15 (47%), with 4–10 les, 12 of 17 (71%) with >10 lesion. EDSS ≥ 6 0 of 21 (0%) with 0 les, 2 of 18 (11%) with 1–3 les, 4 of 15 (27%) with 4–10 les, 9 of 17 (53%) with >10 lesions T2-LV EDSS >3 0 of 21 (0%) with 0 cm^3^, 5 of 18 (28%) with 0.6 cm^3^, 7 of 15 (47%) with 0.9 cm^3^, 12 of 17 (71%) with 5.6 cm^3^ EDSS ≥ 6 0 of 21 (0%) with 0 cm^3^, 2 of 18 (11%) with 0.6 cm^3^, 4 of 15 (27%) with 0.9 cm^3^, 9 of 17 (53%) with 5.6 cm^3^
Sailer et al. (1998) [23]	UCL, London centre-based prospective cohort 1984–1987	CIS	10 y	71 58 analysed	BL: 2.0 (0–74.0) 5 y FU: 7.5 (0–103) 10 y FU: 10.5 (0–105)	BL: 0.43 (0–13.7) 5 y FU:1.84 (0–36.5) 10 y FU: 3.39 (0–88.6)	EDSS > 3 (0–5 y = 1.5 (0–8.5)) EDSS >3 (5–10 y = 0.5 (−1.0–5.0)) (0–10 y = 2 (01–10))
O’Riordan et al. (1998) [22]	UCL, London centre-based prospective cohort 1984–1987	CIS	10 y	129 81 analysed	⦸	⦸	10 y FU EDSS >3 0 of 27 (0%) with 0 les, 0 of 3 (0%) with 1 les, 5 of 16 (31%) with 2–3 les, 4 of 15 (27%) with 4–10 les, 14 of 20 (75%) with >10 les EDSS > 5.5 1 of 27 (4%) with 0 les, 0 of 3 (0%) with 1 les, 2 of 16 (13%) with 2–3 les, 3 of 15 (20%) with 4–10 les, 7 of 20 (35%) with >10 les

* Studies included in the meta-analysis (quantitative). The data are presented as median (range), (mean ± SD) unless otherwise stated; y: years, BL: Baseline, FU: follow-up, ⦸: not report, les: lesion, LV: lesion volume.

## Data Availability

No new data were created or analysed in this study.

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
