# Peer review of "Magnetic Resonance Imaging as a Prognostic Disability Marker in Clinically Isolated Syndrome and Multiple Sclerosis: A Systematic Review and Meta-Analysis"

_diagnostics, 2022, doi:10.3390/diagnostics12020270_

Round 1

Reviewer 1 Report

In this review article, Altokhis et al. performed a meticulous and systematic literature assessment on the prognostic value of MRI-quantified white matter lesions and its effect on long term disability in multiple sclerosis. Strict selection criteria were applied on the original number of identified records (n=1108), so that 15 studies were retained for qualitative analysis and three remained for meta-analysis. Based on these studies, the authors concluded that MRI quantified white matter lesions on T2 MRI were significantly associated with long term disability as measured by the Expanded Disability Status Scale (EDSS). This association critically depended on the number of lesions, with high lesion numbers having higher odds (n>10: OR 5.54 vs n=4: OR 4.3). This review article applied solid methodology and transparent selection criteria. Moreover, it tackles an important clinical question in the multiple sclerosis field. Some minor points could however be improved.

-The Quadas-2 tool was used to evaluate bias. Can the authors elaborate on this method and discuss which potential bias it addresses specifically? Note that in Table 1 potential bias is described by ‘QUIPS’: should this be ‘Quadas-2’?

-In Quality assessment it is mentioned that ‘essential’ data was often missing. What does this imply?

-Tables would benefit from chronological ordering of references.

-Table 2 could benefit from a description of the actual segmentation procedure of white matter lesions: which software package and version, was there manual delineation of lesions performed or based on an automatic process, etc..?

-Given the clinically relevant question, a harmonized procedure for segmentation of lesions could be discussed in the Discussion as this might become important for future studies (E.g. “HyperMapper” (Forooshani et al., 2021), a state-of-the-art automatic segmentation tool that could potentially be useful) or any other harmonization criteria could be addressed.

-Figure 2 and 3 appear confusing due to tables being incorporated as figures: it would be better to only show the two forest plots in 1 figure and the tables separately. A more detailed explanation of the statistics in the legend is advised as well (e.g. Tau2).

Other unclear points regarding the forest plots should be addressed:

Does the size of the symbols have any meaning?

What does the arrow show?

The number of lesion description below the x-axis is a bit confusing.

-If possible, the statistics (OR) can be mentioned in the abstract.

Author Response

Dear Editor,

We are grateful for your consideration of this manuscript, and we also very much appreciate your suggestions, which have been very helpful in improving the manuscript. Thank you to the reviewer for their careful reading of our text.

All the comments we received in this review have been taken into account in improving the quality of the review, and we present our reply to each of them separately (Please see the attachment). Together with the use of (track changes) in the main manuscript. 

Kind regards, 

Amjad 

Reviewer 2 Report

we read with interest the article by Altokhis  et al  entitled Magnetic Resonance Imaging as a Prognostic Disability Marker in Clinically Isolated Syndrome and Multiple Sclerosis: A Systematic Review and Meta-Analysis linking the prognostic value of MRI in the early MS disease to long-term physical disability.

The study is well-designed with a very precise experimental design for conducting this meta-analysis. the read out is excellent

The only things that the author can comment on is:

1- State the inclusion and exclusion criteria for the selection of the cohorts

2-segrgre the MS population among those that are treated with different drug intervention, duration and longtivity vs the non-treated as this can be a confounding factor,

3- Discuss male/female and age in the discussion

Author Response

(The authors gave the same response as above.)
